# An In Vitro Artificial Wound Slough–Biofilm Model Developed for Evaluating a Novel Antibiofilm Technology

**DOI:** 10.3390/microorganisms12112223

**Published:** 2024-11-02

**Authors:** Rui Chen, Jeanne Saint Bezard, Marcus J. Swann, Fergus Watson, Steven L. Percival

**Affiliations:** 5D Health Protection Group Ltd., Liverpool L7 8XZ, UK; rui.chen@5dhpg.com (R.C.); jeanne.saintbezard@5dhpg.com (J.S.B.); marcus.swann@5dhpg.com (M.J.S.)

**Keywords:** artificial wound slough, antibiofilm, chronic wounds, biofilm disruption

## Abstract

Eschar and slough in wounds serve as a reservoir for microorganisms and biofilms, damaged/devitalised cells, and inflammatory chemokines/cytokines, which work to initiate and prolong persistent inflammation and increase the risk of infection. Biofilm-related inflammation and infections are considered to be highly prevalent in acute wounds and chronic wounds. As slough is known to harbour biofilms, measuring the efficacy of antimicrobials in killing microbes both within and under slough is warranted. This highlights the need for more clinically relevant wound biofilm models to address this significant clinical need. Consequently, in this study, we developed an in vitro artificial wound slough (AWS) biofilm model produced by forming a biofilm below a layer of AWS, the latter of which was composed of the main protein components reported in wound eschar and slough, namely collagen, elastin, and fibrin. The model was employed to investigate the antibiofilm and antibacterial efficacy of a new patented smart next-generation antibiofilm technology composed of silver–zinc EDTA complexes and designed as a family of multifunctional metal complexes referred to as MMCs, in a liquid format, and to determine both the performance and penetration through AWS to control and manage biofilms. The results demonstrated the ability of the AWS–biofilm model to be employed for the evaluation of the efficacy of a new antibiofilm and antimicrobial next-generation smart technology. The results also demonstrated the potential for the proprietary EDTA multifunctional metal complexes to be used for the disruption of biofilms, such as those that form in chronic wounds.

## 1. Introduction

Chronic wounds, such as diabetic foot ulcers, venous leg ulcers, and pressure ulcers, are a significant concern in healthcare due to their prolonged healing process, which has placed a significant burden on both the patient and the already overwhelmed healthcare system [1,2]. These types of wounds, which are often stuck in the inflammatory phase of healing, can lead to complications like the formation of eschar and slough. Slough is a soft hydrated yellow or white layer comprising dead and living cells and debris, biofilms, microorganisms and proteinaceous materials, including various types of collagen, fibrin, elastin, matrix metalloproteases, and proteins related to inflammatory immune responses [3]. Layers of slough, which covers the wound, can serve as an adhesive surface for microbial attachment, allowing for the proliferation and production of an extracellular polymeric substance (EPS) which serves as a protective matrix biofilmwhich is very immunogenic. The presence of biofilm in slough can impede wound healing by maintaining a chronic inflammatory state. This leads to elevated levels of tissue-degrading enzymes and reactive oxygen species (ROS), which damage cells and molecules necessary for healing [3,4]. Biofilms within slough are particularly tolerant to antibiotics and the immune system, making infections difficult to treat and manage. Effective chronic wound management often requires the resolution of inflammation by the neutralisation and elimination of the harmful agents residing in slough to expose the biofilm and allow for targeted antimicrobial treatments [3]. Biofilms within slough are most effectively removed with regular and vigorous debridement in the clinic [5,6,7,8,9]. The main methods of debridement include biological debridement, using sterile maggots to digest dead tissue and pathogens [10]; enzymatic debridement, using enzymes such as bromelain to chemically liquefy necrotic tissue [11,12]; autolytic debridement, using the patients’ own enzymes with a wound dressing such as a hydrogel or hydrocolloids to moisten and liquefy necrotic tissue [9]; mechanical debridement, using ‘wet to dry’ gauze to physically remove devitalised tissue; and surgical sharp and conservative sharp debridement, using a scalpel, pair of scissors, or forceps to remove devitalised tissue, usually by a surgical practitioner [5]. Sometimes, an array of debridement methods are applied in complex chronic wound care to improve wound healing and reduce the risk of infection [6]. Each method has its own benefits and drawbacks; for example, surgical debridement is fast and effective but is invasive and often extremely painful, requiring specialist training, and it is relatively expensive so it is not routinely standard practice.

Recently, chemical debridement, which uses special chemical compounds to help facilitate the removal of slough and eschar, has been used as an alternative to surgical debridement. Sometimes, antimicrobials, such as polyhexamethylene biguanide (PHMB), hypochlorous acid, or ionic silver, are added as preservatives to improve the antimicrobial potential of the products [13]. Chemical debridement can enable the removal of slough and biofilm in a simple, safe, and effective manner that can be conducted without special training [11].

There is currently no standard, reproducible and validated method to evaluate the effects of a chemical compound on slough breakdown and any accompanying antimicrobial efficacy. It is key to determine the efficacy of chemical compounds on debridement and antibacterial efficacy in vitro before the development of platforms based on these compounds. In a previous study, we demonstrated an artificial wound model that can be mimetic of wound eschar and slough and which has been successfully applied to analyse the efficacy of debridement [11,14]. In this study, the artificial wound model has been further developed to include the formation of biofilm under the slough layer to produce a new cost-effective artificial wound slough (AWS)–biofilm model using *Pseudomonas aeruginosa*. The validity of the AWS–biofilm model has been demonstrated by measuring silver penetration and antibiofilm efficacy of a multifunctional metal complex (MMC) containing silver.

## 2. Materials and Methods

### 2.1. Wound Treatment Solutions

The two silver-containing solutions used to assess antimicrobial and antibiofilm efficacy were an MMC, which was composed of silver, zinc, and ethylenediaminetetraacetic acid (EDTA) at 3.0% (*w*/*v*), and a silver nitrate solution at 1.1% (*w*/*v*). Both solutions contained equal amounts of silver ions (64.8 mM or 6985 ppm). All components were purchased from Fisher Scientific (Loughborough, UK).

### 2.2. Bacterial Culture

A 24 h biofilm model was produced using a 48-well plate consisting of a *Pseudomonas aeruginosa* (ATCC 700888) culture. The culture was set up by inoculating Tryptone Soy Agar (TSA) (SLS, Nottingham, UK) with working cultures stored at −80 °C and incubated for 24 h at 37 °C in a static incubator. The resultant growth was then suspended in Tryptone Soy Broth (TSB) and adjusted to 10^6^ colony-forming units (CFUs) per mL.

### 2.3. AWS Preparation

Artificial wound slough (AWS) was composed of 65% collagen, powder from bovine Achilles tendon; 10% elastin, powder from bovine neck ligament; and 25% fibrinogen, from bovine plasma as previously reported in the literature [14]. The solution was homogenised using a homogeniser (Stuart^®^ SHM1) for 10 min before being clotted by mixing with thrombin (6.25 U/mL) lyophilised powder from bovine plasma in a 1:1 (*v*/*v*) ratio. All components were purchased from Sigma-Aldrich (Gillingham, UK).

### 2.4. Silver Penetration

Silver ion penetration through the AWS layer was monitored electrochemically by forming the AWS layer directly on top of a screen-printed electrochemical sensor (Figure 1a). The silver that penetrated through the base of the AWS layer to the sensing electrode surface was measured periodically using anodic stripping voltammetry (ASV) throughout the experiment. 

The electrochemical experiment was set up as follows: Short lengths (15 mm) of silicone tubing (internal diameter: 6.5 mm) were bonded on top of printed electrochemical sensors using silicone rubber (Dow Corning RTV 3140) to form a small sample cuvette above the sensor. The printed sensors (Flex-Medical Solutions Ltd., Livingston, UK) consisted of a circular carbon working electrode (diameter: 3 mm), surrounded by a carbon counter electrode and printed Ag/AgCl pseudo reference electrodes. The AWS was formed on top of the sensor by adding the AWS suspension over the sensor and allowing it to cross-link at 37 °C for 30 min, after which the AWS-coated sensors were sealed with parafilm and stored at 4 °C until use within 48 h. AWS layers with thicknesses of 0.0 mm, 0.5 mm, 1.0 mm, and 2.0 mm were produced by adding 0 µL, 15 µL, 30 µL, and 60 µL, respectively, of total AWS solution to the sensor surface. Prior to use, a 6 mm disc of gauze (Multisorb, BSN Medical GMBH, Hamburg, Germany) was placed on top of the AWS layer to reduce the effect of any mechanical disturbance whilst adding test solutions on top.

The anodic stripping profile consisted of holding the working electrode at −0.5 V vs. Ag/AgCl for 50 s to reduce ionic silver from within the layer, followed by a cyclic voltammetry (CV) scan from −0.05 V to 0.9 V and back to −0.05 V at 20 mV/s to re-oxidise (strip) the deposited silver. 

The sensors with AWS layers were first measured at t = 0 h before the addition of the AgNO_3_ and MMC test solutions (200 µL) and then at t = 0.5, 2.5, 6, and 24 h; all tests were performed at room temperature. Sensors without AWS were measured at t = 0 with 100 µL of phosphate-buffered solution (PBS), which was then removed before the addition of the AgNO_3_ and MMC test solutions. The data were analysed by integrating all the current passed during the CV to determine the oxidative charge passed. The charge passed was converted to an equivalent AgNO_3_ concentration value using AgNO_3_ and MMC calibration curves obtained using bare electrodes. The ASV protocol chosen limited the upper dynamic range of the ASV experiment to ~10 mM Ag^+^ so that the lower Ag^+^ concentrations could be measured. 

### 2.5. Antibiofilm Efficacy

The AWS–biofilm model was set up by inoculating each well in a 48-well plate with 400 μL of 10^6^ CFU/mL of *Pseudomonas aeruginosa* and incubated overnight at 37 °C (Figure 1b). At 24 h, the TSB was aspirated from the wells before being washed gently with PBS to remove loosely attached cells. Following this, an AWS layer was added onto the surface of the established biofilm at the following thicknesses of 0.5 mm, 1.0 mm, and 2.0 mm by adding 60 μL, 120 μL, and 240 μL of AWS solution, respectively. The plates were then moved into a shaking incubator set at 37 °C and 30 rpm for 30 min to form the AWS–biofilm in each well.

Once an AWS–biofilm model was generated, 400 µL of the silver test solution was carefully added to the surface and incubated at 37 °C and 30 rpm; for comparison, PBS was used as a negative control. At 2.5 and 24 h, the AWE layer and wound treatment solutions were aspirated from the biofilm surface and the well was rinsed with PBS to remove any remaining residue, before adding 400 µL of neutralising solution to quench further antimicrobial activity. The plates were then sonicated for 30 min, and the neutralising solution/dispersed biofilm from each well was subsequently serially diluted in PBS at a ratio of 1:10 and enumerated by pipetting 100 µL onto TSA plates and incubating overnight at 37 °C.

### 2.6. Confocal Laser Scanning Microscopy (CLSM)

Samples of AWS–biofilm models were subjected to confocal laser scanning microscopy (CLSM) analysis. After the 24 h exposure time to the wound treatment solutions, the AWS layer and the solutions were removed and the wells were rinsed with PBS. The resultant biofilms were stained with the LIVE/DEAD™ Baclight™ fluorescent stains adjusted to a final concentration of 2.5 μM of SYTO 9^®^ and 27.5 μM of propidium iodide before being incubated at room temperature in the dark for 20 min. The stain was then aspirated, and the biofilms were washed carefully with PBS; 20 μL of PBS was subsequently added to each well to ensure the biofilm remained hydrated. The stained biofilms were visualised with an LSM 780 Zeiss confocal microscope with a 20x (0.9 NA) air objective. Samples were imaged in triplicate, and all images were taken under identical conditions and the images were analysed by Zeiss Zen 3.10 (Carl Zeiss Microscopy GmbH, Oberkochen, Germany). The ratio of live bacteria to total bacteria in the biofilm was calculated from the intensity of green fluorescence to total fluorescence intensity. 

### 2.7. Statistical Analysis

All data were presented as the mean ± standard deviation and performed in triplicate unless stated otherwise. The differences were tested for statistical significance using a two-way ANOVA with Tukey’s multiple comparisons or Sidak’s multiple comparisons using Prism 7 software.

## 3. Results

### 3.1. Electrochemically Measured MMC and AgNO_3_ Penetration of AWS

A set of experiments were conducted in the absence of AWS with increasing concentrations of AgNO_3_ and MMC solutions to produce calibration curves to allow the measured amount of Ag^+^ penetration through the AWS layer to be quantified, as shown in Figure 2 below. 

The data from the time series experiment measuring the silver penetration of AgNO_3_ and MMC test solutions through 0.0 mm, 0.5 mm, 1.0 mm, and 2.0 mm thick AWS layers are shown in Figure 3 and values are tabulated in Table 1. For comparison, control data using water as the test solution on 2.0 mm thick AWS layers were also collected. The average measured charge values were converted to nominal silver concentrations using the calibration values from Figure 2. The plots show that after adding the test solutions (t = 0.5 h), a signal above that corresponding to 10 mM Ag^+^ was obtained with both AgNO_3_ and MMC solutions for the blank sensors (0.0 mm AWS) and the 0.5 mm AWS-coated sensors. At 1.0 mm AWS, the response measured for the AgNO_3_ and MMC solutions differed, with the former showing less than a quarter of the value of the MMC solutions on average (1.8 mM vs. 8.3 mM, respectively) with no penetration observed for two of the AgNO_3_ samples. After 2.5 h for 1.0 mm AWS, penetration is seen on all samples and both solutions have stabilised and do not increase further, with the MMC signal remaining approximately four times that of the AgNO_3_ solution.

The response for the 2.0 mm thick AWS layers was dramatically reduced in comparison to the thinner layers over time, with no detectable increase in signal for either the AgNO_3_ or MMC solutions half an hour after their addition. After 2.5 h, silver penetration can be observed, with the signal from the MMC over an order of magnitude greater than that from AgNO_3_ (0.15 mM vs. 0.007 mM, respectively). The MMC penetration continues to increase until stabilising at 6 h at 0.38 mM, a lower value than that observed for the thinner AWS layers (~10 mM). The average AgNO_3_ response measured continued to increase beyond 24 h; however, it remained below that of the equivalent MMC data. For two of the samples at 2.0 mm using AgNO_3_, no change in response was observed after 24 h.

The results indicate that, for thinner AWS layers, the majority of the silver penetration occured within the first 0.5 h with little variation in the following time points, and that the thickness of the AWS had an indirect correlation. It is also clear that the MMC test solution achieved greater silver penetration in thicker AWS layers compared with AgNO_3_, which in turn had a high degree of variability as shown by the large standard deviation.

### 3.2. Antibiofilm Efficacy of MMC and AgNO_3_ Test Solutions on AWS–Biofilm Model

The AWS–biofilm model comprised a *P. aeruginosa* biofilm situated beneath varying thicknesses of AWS layers. The *P. aeruginosa* biofilm prior to the AWS being added exhibited an average cell density of 8.93 ± 0.02 Log_10_ CFU/mL; the addition of the AWS layers, 0.5 mm, 1.0 mm, and 2.0 mm, had no impact on the viability of beneath the biofilm at 0 h as shown in Table 2 and demonstrated microbial bioburden levels between 8 and 9 log after 2.5 and 24 h contact with the negative control. In contrast, after 2.5 or 24 h exposure to either wound treatment solutions, the 0.5 mm thick AWS–biofilm models exhibited complete inactivation. For the 1 mm thick AWS layers, only the MMC demonstrated total microbial kill in the biofilm, with AgNO_3_ demonstrating a 6 log reduction; however, both solutions showed complete eradication of the microbes within the biofilm after a contact time of 24 h. For 2.0 mm AWS thicknesses, reduced levels of inactivation were observed at 2.5 h with the silver nitrate achieving a ~2 log reduction and the MMC achieving a ~5 log reduction; however, at 24 h, the MMC solution was able to achieve complete kill of all microbes within the biofilm, whilst the AgNO_3_ solution exhibited ~6 log reduction. It is clearly demonstrated that the MMC solution exhibited an enhanced and synergistic antibiofilm effect when compared to silver alone. This was most noticeable at the 24 h time point with the MMC solution being significantly more potent than silver nitrate alone.

### 3.3. Antibiofilm Efficacy Analysis by Confocal Microscopy

After 24 h of treatment, the AWS was gently removed from the well, then the well was washed with PBS and stained with the LIVE/DEAD™ Baclight™ fluorescent staining kit (Figure 4). The top row shows 3D images of 0.5 mm AWS–biofilm; the middle row shows 3D images of 1.0 mm AWS–biofilm; and the bottom row shows 3D images of 2.0 mm AWS–biofilm. The negative control (a, d, and g) demonstrates an abundance of live/viable bacterial cells in the biofilm, as indicated by the green signal, forming microcolonies across the surface, whilst the treated samples (b, e, and h treated with 1.1% AgNO_3_; e, f, and i treated with 3% Ag-Zn MMC) reveal a significant reduction in cell viability and a notable reduction in biofilm coverage as indicated by the red signal, indicating extensive biofilm disruption.

The ratio of live bacteria to total bacteria in the biofilm is shown in Figure 5. The data show that the viability of bacteria decreased significantly after treatments with either AgNO_3_ or Ag-Zn MMC for all three thickness AWS–biofilm models for 24 h. The *P. aeruginosa* viability decreased to 4.3 ± 0.8%, 5.8 ± 1.3%, and 7.6 ± 1.8% after AgNO_3_ treatment for the 0.5 mm, 1.0 mm, and 2.0 mm AWS biofilms, respectively, whilst for MMC treatment, viability decreased more to 1.4 ± 1.2%, 2.3 ± 0.7%, and 4.4 ± 0.9% for 0.5 mm, 1.0 mm, and 2.0 mm AWS–biofilms, respectively. A significantly higher antibacterial efficacy was observed for the 3.0% Ag-Zn MMC treatment than for the 1.1% AgNO_3_ treatment at every thickness level of the AWS–biofilm model tested (*p* < 0.01).

## 4. Discussion

Chronic wounds are those that fail to progress through the normal stages of healing, often due to underlying conditions like diabetes or poor supply of nutrients [1,4]. These wounds can be complicated by the presence of biofilms and slough. Biofilms play a significant role in the persistence and chronicity of wounds, and effective treatment involves a combination of debridement, removal of dead tissue and biofilm, antimicrobial therapies, and maintenance of a balanced wound environment, such as exudate or pH levels [15]. Ongoing research aims to develop innovative strategies for managing biofilms in chronic wounds and this includes new antimicrobial agents, biofilm-disrupting technologies, and improved diagnostic methods. In vitro wound biofilm models offer a controlled environment to study biofilm formation and eradication processes outside of the patient. These models not only allow researchers to investigate cellular and molecular mechanisms of biofilm formation with precision, reducing the complexity and variability inherent within in vivo studies, but they are also valuable in the development of new antibiofilm technologies, enabling the screening of potential chemicals for efficacy and toxicity before proceeding to animal or clinical trials. Traditionally, in vitro biofilm models which include microtiter plate assays [16], flow cells [17], constant depth film fermenters [18], annular reactors [19,20], and perfused biofilm fermenters [20,21] are widely used to advance our understanding of biofilms and develop effective strategies to control them in medical, industrial, and environmental settings. However, there is no standard or specific chronic wound biofilm model that considers the complications of slough, which enhances the attachment and protection for biofilm in the wound bed.

In this study, we reported a novel cost-effective artificial wound slough biofilm model as a mimetic clinical wound environment and its potential application in determining the ability of a wound dressing, wound irrigation solution, or key components of such to enhance slough penetration to execute its antibiofilm potential. The model involves forming a biofilm below the layer of AWS, comprising the main protein components in wound eschar and slough: collagen, elastin and fibrin. The AWS layer can also be formed directly onto an EC sensor which enables us to independently evaluate the integrity and penetrative characteristics of the AWS layer after treatment by measurement of the active substances in the treatment as they penetrate through the AWS layer to reach the EC sensor surface.

In the study, the AWS–biofilm model was tested with two liquids: a new antibiofilm technology (patent-protected) comprising 3.0% Ag-Zn MMC and a solution of 1.1% AgNO_3_, selected to have the same level of silver (64.8 mM) as a suitable positive control. The gradient measured for the EC AgNO_3_ calibration response is slightly higher than that for the MMC, which is most likely because the silver in the MMC is partially coordinated by EDTA and so it had a lower solution diffusion coefficient. For the AWS model, the AWS layers contain PBS as well as the protein components, so the exact solution ion composition is unclear. This means that the calibrations can only be an approximation. It should also be noted that the presence of the AWS layer on the electrode surface is likely to reduce the magnitude of the Ag^+^ signal, so this is another reason that it does not represent an exact calibration of the Ag^+^ concentration at the electrode surface beneath the AWS layer, but it is none the less indicative of the timescale and extent of the Ag^+^ penetration through the AWS layer.

Whilst the protocols of the EC and microbiological assays are slightly different, such as a static experiment at room temperature for EC compared with an experiment at 37 °C with shaking for the biofilm assay, comparing the EC measurements to the microbiological biofilm cell count assay shows a striking correlation. For example, at 2.5 h, where total kill is not achieved for the 2.0 mm AWS layers, the EC data also show lower levels of silver. This can be seen for AgNO_3_ at the thinner AWS thickness, also; and the least amount of silver penetration was observed for the 2.0 mm layer, which also showed the highest residual bacterial cell density. The EC and microbiological data correlation observed here also agrees well with data that has been previously [22], with *P. aeruginosa* biofilms, looking at silver efficacy in simulated wound fluid probed with an electrochemical wound dressing sensor. This showed that the available Ag^+^ as measured electrochemically needed to be above 1.5 ppm (0.014 mM) for antibiofilm efficacy to be observed. In this work for AgNO_3_ on 2 mm AWS at 2.5 h, we measured 0.007 ± 0.008 mM (0.76 ± 0.86 ppm) and observed only a small (2.3 Log CFU/mL) reduction in the AWS–biofilm.

All the results showed that silver ions penetrated through the AWS to reach the biofilm underneath. However, some silver was evidently trapped within the AWS membrane as a very dark hue was observed on the AWS membrane by the end of the experiment due to the reaction of the silver with the protein in the AWS layer. The addition of AgNO_3_ or MMC solutions to the AWS layers in the measurement vials and well plates caused several processes to occur [22]. As the silver, plus other ions and components, from the solutions start to penetrate the AWS layer, the free silver ions in the solution will precipitate with chloride and phosphate ions and will also bind to and react in the presence of proteins. The data show a faster and greater extent of penetration of the AWS layer by the silver from the MMC solution compared to that in the AgNO_3_ solution. The presence of MMC is likely to modify the ingress of silver into the AWS layer, as EDTA will reduce the amount that precipitates to form silver chloride and may, together with zinc ions present in the MMC, also compete with silver binding to proteins in the AWS [22]. This will increase the silver that remains available in solution and is able to diffuse through the AWS layer. After 24 h, the signals were stable and the surfaces of the AWS layers were dark from the reaction of silver.

In recent years, major advances have been made to make complex in vitro chronic wound models [23,24,25]. Most of these models focused on in vitro culture and characterising chronic wound-derived fibroblasts, which exhibited cellular pathological morphology in cell shape, cell proliferation and migration ability, and cell secretions [26]. The models provided a microenvironment mimic to investigate the cellular and molecular mechanism of wound healing. Several models discussed the infection and biofilm formation in chronic wound healing [27,28]. However, some key factors critically involved in chronic wound healing, such as eschar and slough, which is a hallmark of chronic wounds, were absent from these models. The results of this study demonstrate that the AWS–biofilm model can be used to mimic clinically chronic wounds where eschar or slough has formed and impedes wound healing and can be used to investigate if antimicrobials are able to be truly effective therapeutically. The AWS–biofilm data were corroborated by EC measurements of the time-dependent silver penetration through the AWS layers. However, the AWS–biofilm model has inherent limitations typical of in vitro models. Wound slough is primarily composed of proteins associated with the structure and formation of the skin, blood clot formation, and various immune responses [1,3]. Slough is also a reservoir of microorganisms and biofilms with highly polymicrobial signatures associated with both wound aetiology and locations [4]. This composition can vary depending on the wound type and the patient’s overall health [29]. To reduce the complexity and improve standardisation, our model was mainly based on three main proteins, collagen, elastin, and fibrin, which are found in wound eschar and slough, and as a mono-species biofilm. As such, the model represents a greatly simplified wound environment: lacking nutrient and oxygen gradients, showing an absence of cell and immune response, and demonstrating limited species interactions. Future iterations of the AWS–biofilm model are on-going at and involve the use of multispecies biofilms as well as ex vivo skin samples to begin to address some of these limitations of this and other in vitro models which will to help accelerate new antimicrobial and antibiofilm discovery, enhance clinical relevance and improve patient outcomes.

## Figures and Tables

**Figure 1 microorganisms-12-02223-f001:**
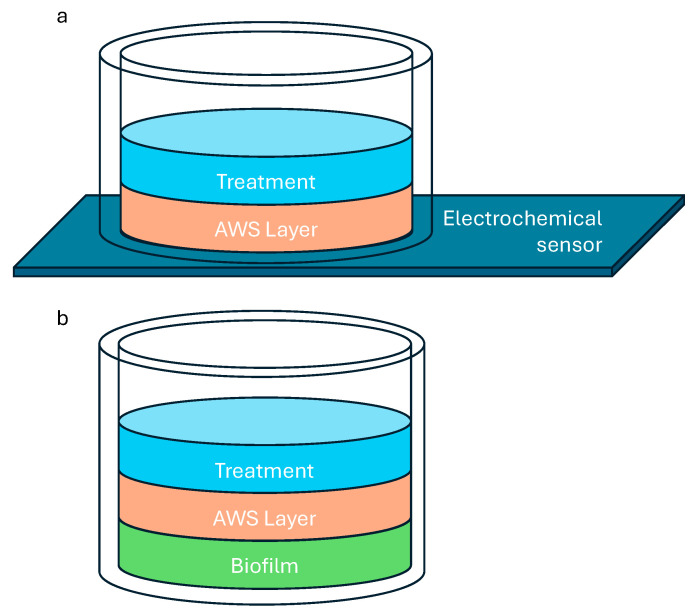
A schematic of the experimental design for (**a**) the electrochemical sensor and (**b**) the AWS–biofilm model.

**Figure 2 microorganisms-12-02223-f002:**
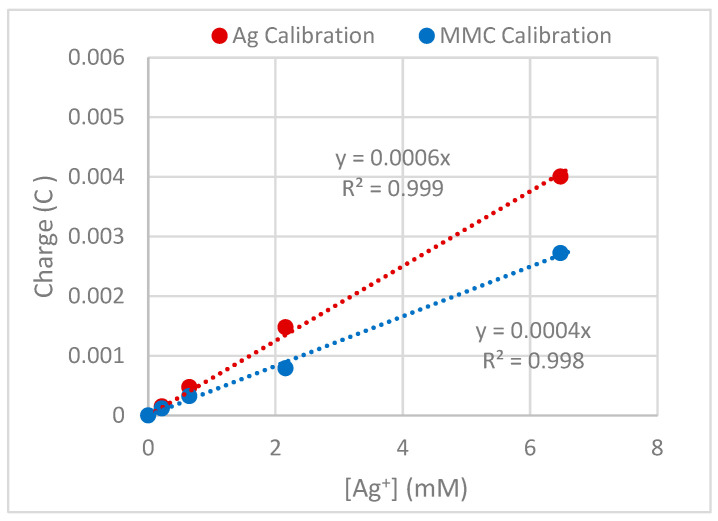
Charge vs. AgNO_3_ and MMC silver concentration (mM) plot from anodic stripping voltammetry experiment for bare sensor electrode.

**Figure 3 microorganisms-12-02223-f003:**
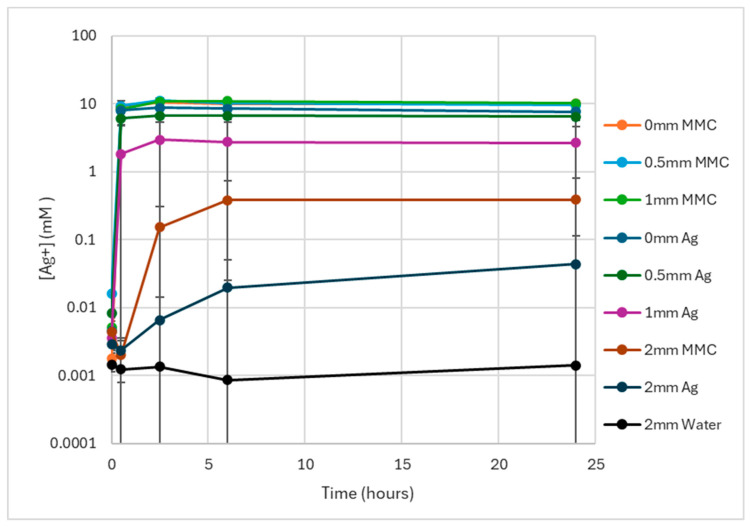
Silver penetration vs. time (hours) with the silver concentration determined using the calibration curves in Figure 2. Curves correspond to samples with 0.0 mm, 0.5 mm, 1.0 mm, and 2.0 mm thick AWS samples with 1.1% AgNO_3_ and 3% MMC test solutions (n = 3). The control data using water as a test solution at 2.0 mm are included.

**Figure 4 microorganisms-12-02223-f004:**
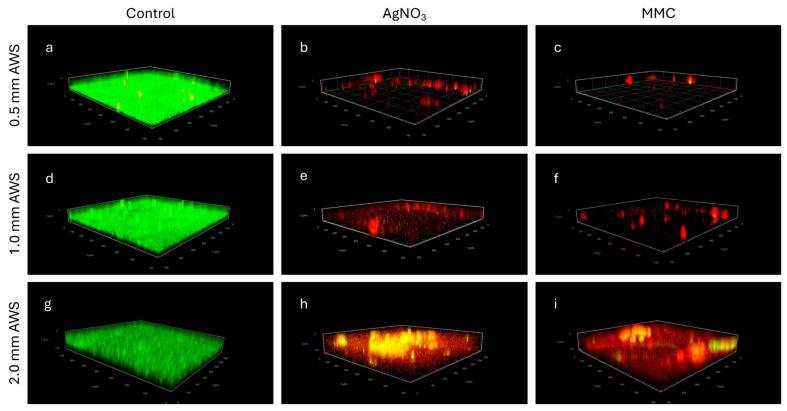
The 3D representative images of the biofilm disruption and inactivation of *P. aeruginosa* biofilm. (**a**–**c**): biofilm under 0.5 mm AWS; (**d**–**f**): biofilm under 1.0 mm AWS; (**g**–**i**): biofilm under 2.00 mm AWS. (**a**,**d**,**g**): control; (**b**,**e**,**h**): treatment with AgNO_3_ solution; (**c**,**f**,**i**): treatment with MMC solution. Biofilm cells were stained by using LIVE/DEAD™ BacLight™ bacterial viability kit which contains SYTO 9 (green fluorescence, viable cells) and propidium iodide (red fluorescence, nonviable cells).

**Figure 5 microorganisms-12-02223-f005:**
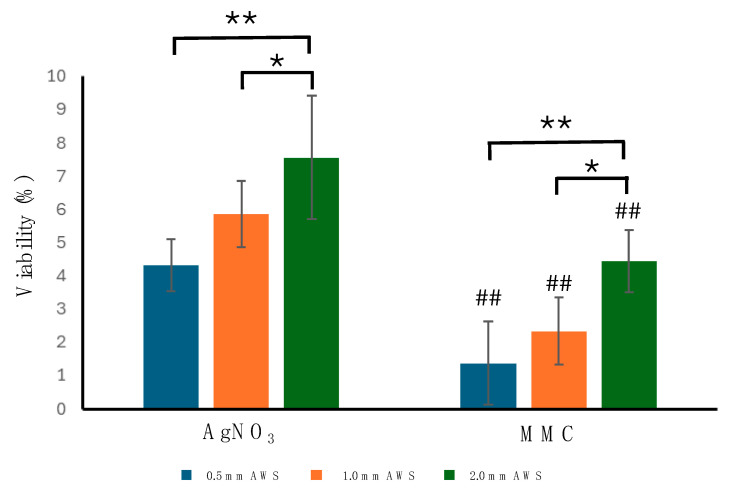
Corresponding quantitative analysis of bacterial viability calculated by the green/total fluorescence ratio counts. Data are shown as mean ± standard deviation. Analysis of variance by Tukey’s multiple comparisons indicated that 2.0 mm AWS layers had significantly greater levels of viability than 0.5 mm and 1.0 mm (*: *p* < 0.05; **: *p* < 0.01). Additionally, using Sidak’s multiple comparisons for testing significance, a statistically significant difference between AgNO_3_ and MMC solutions for each thickness level of AWS was observed (##: *p* < 0.01).

**Table 1 microorganisms-12-02223-t001:** The silver penetration given as an average Ag^+^ concentration (mM) at the sensor surface for MMC and AgNO_3_ solutions. The errors represent the standard deviation of the mean.

**AWS Layer (mm)**	**MMC Solution**				
	**Silver Concentration (mM) at:**				
	**0 h**	**0.5 h**	**2.5 h**	**6 h**	**24 h**
0.0	0.002 ± 0.001	8.800 ± 0.392	10.540 ± 0.696	9.953 ± 0.594	10.018 ± 0.650
0.5	0.016 ± 0.011	9.276 ± 0.106	10.983 ± 0.133	10.255 ± 0.177	9.498 ± 0.074
1.0	0.005 ± 0.003	8.328 ± 0.308	10.788 ± 0.325	10.809 ± 0.197	10.162 ± 0.899
2.0	0.004 ± 0.002	0.002 ± 0.001	0.151 ± 0.152	0.378 ± 0.353	0.385 ± 0.409
**AWS Layer (mm)**	**AgNO_3_ Solution**				
	**Silver Concentration (mM) at:**				
	**0 h**	**0.5 h**	**2.5 h**	**6 h**	**24 h**
0.0	0.005 ± 0.002	7.904 ± 0.164	8.713 ± 0.377	8.455 ± 0.015	7.569 ± 0.158
0.5	0.008 ± 0.001	6.051 ± 1.164	6.631 ± 0.444	6.636 ± 0.200	6.446 ± 0.296
1.0	0.004 ± 0.001	1.806 ± 3.123	2.934 ± 3.372	2.706 ± 3.100	2.649 ± 2.947
2.0	0.003 ± 0.002	0.002 ± 0.001	0.007 ± 0.008	0.020 ± 0.031	0.043 ± 0.071

**Table 2 microorganisms-12-02223-t002:** The average cell density (Log_10_ CFU/mL) for *P. aeruginosa* AWS–biofilms after treatment with AgNO_3_ and MMC test solutions. The error represents the standard deviation of the mean and the limit of detection was indicated at 2 Log_10_.

Test Solutions	0.5 mm AWS			1.0 mm AWS			2.0 mm AWS		
	0 h	2.5 h	24 h	0 h	2.5 h	24 h	0 h	2.5 h	24 h
Control	9.06 ± 0.02	9.19 ± 0.28	9.24 ± 0.08	8.97 ± 0.04	8.91 ± 0.15	9.15 ± 0.08	8.99 ± 0.07	8.09 ± 0.05	9.27 ± 0.02
AgNO_3_		0.00 ± 0.00	0.00 ± 0.00		3.19 ± 0.30	0.00 ± 0.00		5.78 ± 0.16	3.18 ± 0.00
MMC		0.00 ± 0.00	0.00 ± 0.00		0.00 ± 0.00	0.00 ± 0.00		3.01 ± 0.21	0.00 ± 0.00

## Data Availability

The original contributions presented in the study are included in the article, further inquiries can be directed to the corresponding author.

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
