# Peer review of "An In Vitro Artificial Wound Slough–Biofilm Model Developed for Evaluating a Novel Antibiofilm Technology"

_microorganisms, 2024, doi:10.3390/microorganisms12112223_

Round 1

Reviewer 1 Report

Comments and Suggestions for Authors

Dear authors,

An in vitro artificial wound slough – biofilm model developed for evaluating a new smartly triggered next generation anti-biofilm technology

This manuscript entails an informative study focused on preclinical method development and antibacterial efficacy in an area where more complex preclinical models are scarce today. This clearly motivates for investigating a new combined slough and biofilm model based on a previous artificial wound slough (AWS) model developed and published in an article in IWJ 2019 (ref 14), which has now been further established with increased complexity introducing formation of biofilm under the AWS layer. The suggested artificial wound slough-biofilm combined model was utilised to study antibiofilm efficacy and penetration of silver ions either from AgNO3 or a multifunctional metal complex (MMC) entailing silver, zink and EDTA.

Please see below for comments, need of additional information, clarifications or questions:

Materials & Methods:

Regarding describing the creation of the AWS-biofilm model:

-          It could be valuable to describe the reasoning of the percentage chosen of the different components of the AWS utilised (collagen, elastin and fibrin/fibrinogen) connected to the sentence on row 92. Understand this connects to an earlier publication where the slough model was presented, but since the major and vast components of slough in a wound can vary, it could be important to hear your reasoning on the composition as well as the choices taken to create a representative and yet functional preclinical model. You come back to this in a great way in the discussion section on row 337 and onwards where you in a clear manner puts the model into perspective with both strengths and limitations.

-          Since the study presents model development of a new artificial wound slough – biofilm model it might be beneficial for the reader to have a description, e.g. figure/picture for further understanding of the construction. (If room allows)

Materials & Methods & Results:

Regarding 2.5 & 3.2: Antibiofilm efficacy

-          The model described have many interesting potential designs to elucidate different scientific questions in relation to both MMC and AgNO3 penetration of AWS as well as antibiofilm efficacy. It would be interesting to know why the biofilm is first established in the well/surface and then adding the AWS on top instead of creating the biofilm within the AWS layer creating a 3D matrix. In regards of connecting to the distance for silver ions to travel (representing bacterial aggregated below slough in tissue) this makes sense of course, but to mimic the wound tissue, the bacterial component would be found within the slough as well. It of course depends on the purpose but could be an interesting aspect of the model per se to take into consideration as well as the type of biofilm formed.

-          Have you done any analysis of the number of bacteria found within the removed AWS layer after the different time points? How easily can they be removed after the different time points? Further, can and will bacteria invade the AWS part of the model? Slough/Eschar is also composed of bacteria/ aggregates of biofilm, would this model enable visualisation of that?

-          Impact of degradation over 24h: One could speculate that the bacteria forming the biofilm in the bottom of the well could influence the structure of the AWS over at least the 24h period by e.g., degradation. Have you done any observations connected to degradation that might impact the model and thereby interpretation of the data produced?

-          In line with above: the study has a well-defined focus and scope; as a next step it would also be interesting to investigate the integrity of the AWS-biofilm model beyond 24 h. When thinking of clinical practice, dressings are seldom changed every day.

Regarding Table 1:

-          The second heading in relation to AWS Layer (mm) for AgNO3 is not in bold.

Regarding 2.6 & 3.3: Antibiofilm efficacy analysis by confocal laser scanning microscopy

-          How many samples where utilised in this experimental setup?

-          Were there any indications that the difference in thickness (0.5, 1 and 2 mm) of the AWS influenced the P. aeruginosa biofilm structure in any way visualised in the confocal images seen in figure 3 a, d, g? There is limited data presented in relation to the confocal images, besides the total bacteria ratio.

Discussion & Heading:

-          Row 292: The wording in e.g. this sentence could be slightly altered to a more neutral, since the data presented in the paper does not account for the full statement. If not, then e.g.  the wording “smartly triggered” seen in both the heading, abstract and e.g. row 292 would need more explanation in the text to motivate the formulation including references. The available information right now is e.g. row 76 “by measuring silver penetration and antibiofilm efficacy of a multifunctional metal complex (MMC) containing silver.” and rows 295-296 & 327-329 describing  actions of EDTA that could be further demonstrated or referenced.

Author Response

Please see below for comments, need of additional information, clarifications or questions:

Materials & Methods:

Regarding describing the creation of the AWS-biofilm model:

Comments: -          It could be valuable to describe the reasoning of the percentage chosen of the different components of the AWS utilised (collagen, elastin and fibrin/fibrinogen) connected to the sentence on row 92. Understand this connects to an earlier publication where the slough model was presented, but since the major and vast components of slough in a wound can vary, it could be important to hear your reasoning on the composition as well as the choices taken to create a representative and yet functional preclinical model. You come back to this in a great way in the discussion section on row 337 and onwards where you in a clear manner puts the model into perspective with both strengths and limitations.

Response: Thank you for the appreciation regarding the importance of the composition; we believe that there is sufficient explanation within the manuscript as to regards the use of the model – as noted by yourself within the discussion section; however, we have emphasized this further with an additional statement to highlight its origin and the purpose to standardizing the model.

Comments: -          Since the study presents model development of a new artificial wound slough – biofilm model it might be beneficial for the reader to have a description, e.g. figure/picture for further understanding of the construction. (If room allows)

 Response: An additional figure has been included to illustrate the experimental design.

Materials & Methods & Results:

Regarding 2.5 & 3.2: Antibiofilm efficacy

Comments:-          The model described have many interesting potential designs to elucidate different scientific questions in relation to both MMC and AgNO3 penetration of AWS as well as antibiofilm efficacy. It would be interesting to know why the biofilm is first established in the well/surface and then adding the AWS on top instead of creating the biofilm within the AWS layer creating a 3D matrix. In regards of connecting to the distance for silver ions to travel (representing bacterial aggregated below slough in tissue) this makes sense of course, but to mimic the wound tissue, the bacterial component would be found within the slough as well. It of course depends on the purpose but could be an interesting aspect of the model per se to take into consideration as well as the type of biofilm formed.

Comments:-          Have you done any analysis of the number of bacteria found within the removed AWS layer after the different time points? How easily can they be removed after the different time points? Further, can and will bacteria invade the AWS part of the model? Slough/Eschar is also composed of bacteria/ aggregates of biofilm, would this model enable visualisation of that?

Response: Thank you for your perspective on this, we agree that the bacteria are likely to penetrate the AWS layer. This would constitute a fascinating area for study, which is quite independent of any consideration of antimicrobial activity of a wound treatment; unfortunately, this was not within the scope of the study. We believe there are suitable controls, for comparative purposes, in place which allow the identification of antimicrobial activity on surface bound bacteria i.e., biofilm by the treatments. Future iterations could look into this further.

Comments:-          Impact of degradation over 24h: One could speculate that the bacteria forming the biofilm in the bottom of the well could influence the structure of the AWS over at least the 24h period by e.g., degradation. Have you done any observations connected to degradation that might impact the model and thereby interpretation of the data produced?

Response: Thank you for your perspective on this, and we agree that the bacteria would be expected to affect the AWS layer. By removing the AWS layer before quantification and by focusing on the extent to which the underlying biofilm was killed, we sought to remove some of this complexity and use the model to provide a simple quantification of the effect of the antimicrobial on the underlying biofilm. Unfortunately, studying the effect of the biofilm on the AWS layer was not within the scope of the study. We believe there are suitable controls, for comparative purposes, in place which allow the identification of antimicrobial activity on surface bound bacteria i.e., biofilm by the treatments. Future iterations could look into this further.

Comments:-          In line with above: the study has a well-defined focus and scope; as a next step it would also be interesting to investigate the integrity of the AWS-biofilm model beyond 24 h. When thinking of clinical practice, dressings are seldom changed every day.

Regarding Table 1:

Comments:-          The second heading in relation to AWS Layer (mm) for AgNO3 is not in bold.

Response: Corrected.

Regarding 2.6 & 3.3: Antibiofilm efficacy analysis by confocal laser scanning microscopy

Comments:-          How many samples where utilised in this experimental setup? 

Response: In the study, three samples were characterised  per AWS layer and per treatment was used for the microscopy. This has been clarified in the method section 2.6.

Comments:-          Were there any indications that the difference in thickness (0.5, 1 and 2 mm) of the AWS influenced the P. aeruginosa biofilm structure in any way visualised in the confocal images seen in figure 3 a, d, g? There is limited data presented in relation to the confocal images, besides the total bacteria ratio.

 Response: Thank you for your perspective, and whilst subjectively there was no visual impact on the biofilm structure due to limitations within the study were unable to confirm this. We are satisfied that the controls still effectively demonstrate the comparison between treated and untreated; and future iterations should consider this factor.

Discussion & Heading: 

Comments:-          Row 292: The wording in e.g. this sentence could be slightly altered to a more neutral, since the data presented in the paper does not account for the full statement. If not, then e.g.  the wording “smartly triggered” seen in both the heading, abstract and e.g. row 292 would need more explanation in the text to motivate the formulation including references. The available information right now is e.g. row 76 “by measuring silver penetration and antibiofilm efficacy of a multifunctional metal complex (MMC) containing silver.” and rows 295-296 & 327-329 describing  actions of EDTA that could be further demonstrated or referenced. 

Response: Thank you for highlighting this. We have revised the wording to replace ‘smartly triggered’ with novel and included a statement within the introduction and discussion to emphasize it’s novelty.

Reviewer 2 Report

Comments and Suggestions for Authors

This study focused on the development of an in vitro model to mimic the conditions of wound eschar and slough, which can harbor harmful microorganisms and biofilms. The researchers aimed to evaluate a new antibiofilm technology using this model. The results showed that the model effectively simulated the wound environment, and that the new technology had promising potential for disrupting biofilms, potentially improving wound healing and reducing the risk of infection. It's an interesting topic given the significant issue of chronic wounds. However, I’ve some considerations and I’m going to take account again this manuscript after minor revision.

1)     Abstract is well done.

2)     Introduction is well done; however, the aim should be described better.
What’s the objective? To produce a new cost-effective artificial wound slough (AWS) - biofilm model or to test multifunctional metal complex (MMC) containing silver? I suggest defining the aim, specifying the origin of biofilm (from Pseudomonas aeruginosa only because biofilm can derive from multispecies yet). Additionally, Line 33-40: sentences lack references. Line 63-66: sentences lack references.

3)     Methods were well described.

4)     Results are supported by experimental techniques, and by clear images. I suggest inserting some pictures of the wound biofilm model.

5)     Discussion was well argued. I suggest to the authors inserting some studies which focused on in vitro chronic wound models to compare their wound model.

I appreciate that the authors described the limitations of their wound model. However, I suggest arguing the absence of cells such as fibroblasts that have a critical role in the wound healing.

Overall, this paper has a good quality. It has an interesting subject considering that the management of chronic wounds is critical. The manuscript is well written in each part and easily accessible for readers. The choice of experiments was in line with the purposes, and the data are supported by a good statistical analysis. The results are supported by experimental techniques, by clear images. It’s my opinion this manuscript is suitable for the publication after a minor revision.

Author Response

Comments:-          2)     Introduction is well done; however, the aim should be described better. 
What’s the objective? To produce a new cost-effective artificial wound slough (AWS) - biofilm model or to test multifunctional metal complex (MMC) containing silver? I suggest defining the aim, specifying the origin of biofilm (from Pseudomonas aeruginosa only because biofilm can derive from multispecies yet). Additionally, Line 33-40: sentences lack references. Line 63-66: sentences lack references.

 Response: Thank you for your input into this section. We have revised the aim to account for more specificity within the model during the introduction. Additionally, the aforementioned lines have been correctly referenced.

3)     Methods were well described. 

Comments:-          4)     Results are supported by experimental techniques, and by clear images. I suggest inserting some pictures of the wound biofilm model.

 Response: Figure included.

Comments:-          5)     Discussion was well argued. I suggest to the authors inserting some studies which focused on in vitro chronic wound models to compare their wound model.

I appreciate that the authors described the limitations of their wound model. However, I suggest arguing the absence of cells such as fibroblasts that have a critical role in the wound healing. 

Response: Thank you for the suggestion. As advised, we have revised the final paragraph to discuss the other in vitro chronic wound models published in recent years and to compare those with our model.

Reviewer 3 Report

Comments and Suggestions for Authors

The manuscript is well put together. Easy to follow. All the right controls were used.

Author Response

Not applicable

Reviewer 4 Report

Comments and Suggestions for Authors

The paper by Rui Chen et al describes new an in vitro artificial biofilm model developed for evaluating a new smartly triggered next generation anti-biofilm technology. Taking into account the understanding of the impact of biofilms on bacterial susceptibility to antimicrobials,  this topic opens new inside in development of antibiofilm agents and is very important for the clinical practice.

The work is well written and scientifically sounds and can be considered for publication. Nevertheless, some issues should be addressed.

Since the model used to develop the biofilm on AWS is new, this is recommended to make a figure illustrating the experimental design. 

What is the rationale to use only silver-based treatment? While the Ag ions can be easily detected, why do not use any conventional antimicrobial treatment else to demonstrate the efficiency?

Fig 1 - re-write values in equations as y=0.0004x, R2=99%

Fig 3 - rows and columns should be marked on the left and on the top, respectively. 

Comments on the Quality of English Language

There are a number of typos 

Author Response

Comments:-          Since the model used to develop the biofilm on AWS is new, this is recommended to make a figure illustrating the experimental design. 

 Response: Figure included.

Comments:-          What is the rationale to use only silver-based treatment? While the Ag ions can be easily detected, why do not use any conventional antimicrobial treatment else to demonstrate the efficiency?

Response: It would indeed be possible to use other antimicrobial treatments to demonstrate efficacy, however as this is the first publication using this model we felt that using a positive control that could be electrochemically quantified in exactly the same way for both treatments (as is acknowledged by the reviewer) was particularly important to avoid unnecessary complexity in  terms of validation. We have modified the text to make this more evident.     

Comments:-          Fig 1 - re-write values in equations as y=0.0004x, R2=99%

Response: This has been addressed and revised.

Comments:-          Fig 3 - rows and columns should be marked on the left and on the top, respectively. 

Response: This has been addressed and revised.

Comments:-          Comments on the Quality of English Language

There are a number of typos 

Response: We have updated the manuscript to remove typos and improve the readability.